# Generative Adversarial Networks in Dermatology: A Narrative Review of Current Applications, Challenges, and Future Perspectives

**DOI:** 10.3390/bioengineering12101113

**Published:** 2025-10-16

**Authors:** Rosa Maria Izu-Belloso, Rafael Ibarrola-Altuna, Alex Rodriguez-Alonso

**Affiliations:** 1Medicine Faculty, Hospital Universitario Basurto, 48013 Bilbao, Spain; 2Hospital Universitario Galdakao, 48960 Bizkaia, Spain; rafael.ibarrolaaltuna@osakidetza.eus; 3Facultad de Medicina, Universidad del Pais Vasco/EHU, 48940 Leioa, Spain; alecai.rodriguez@gmail.com

**Keywords:** GAN, dermatology, synthetic images, artificial intelligence, StyleGAN, deep learning, medical imaging, skin lesions, segmentation

## Abstract

Generative Adversarial Networks (GANs) have emerged as powerful tools in artificial intelligence (AI) with growing relevance in medical imaging. In dermatology, GANs are revolutionizing image analysis, enabling synthetic image generation, data augmentation, color standardization, and improved diagnostic model training. This narrative review explores the landscape of GAN applications in dermatology, systematically analyzing 27 key studies and identifying 11 main clinical use cases. These range from the synthesis of under-represented skin phenotypes to segmentation, denoising, and super-resolution imaging. The review also examines the commercial implementations of GAN-based solutions relevant to practicing dermatologists. We present a comparative summary of GAN architectures, including DCGAN, cGAN, StyleGAN, CycleGAN, and advanced hybrids. We analyze technical metrics used to evaluate performance—such as Fréchet Inception Distance (FID), SSIM, Inception Score, and Dice Coefficient—and discuss challenges like data imbalance, overfitting, and the lack of clinical validation. Additionally, we review ethical concerns and regulatory limitations. Our findings highlight the transformative potential of GANs in dermatology while emphasizing the need for standardized protocols and rigorous validation. While early results are promising, few models have yet reached real-world clinical integration. The democratization of AI tools and open-access datasets are pivotal to ensure equitable dermatologic care across diverse populations. This review serves as a comprehensive resource for dermatologists, researchers, and developers interested in applying GANs in dermatological practice and research. Future directions include multimodal integration, clinical trials, and explainable GANs to facilitate adoption in daily clinical workflows.

## 1. Introduction

Artificial Intelligence (AI) is revolutionizing modern medicine by offering advanced tools for the detection, diagnosis, and management of numerous diseases [1,2,3]. Dermatology, in particular, has greatly benefited from the Deep Learning (DL) techniques applied to skin image analysis, enabling automated classification, lesion segmentation, and the development of predictive models for a wide range of dermatoses [4,5].

Among DL methods, Generative Adversarial Networks (GANs) have shown remarkable potential in dermatology [6]. These networks are capable of generating high-fidelity synthetic skin images, addressing critical limitations in current datasets. For example, only 12% of existing dermatological image repositories include phototypes IV–VI, contributing to diagnostic biases in non-Caucasian populations [7]. GANs can augment these under-represented categories, improve image quality, simulate rare or atypical lesions, and support data anonymization—key aspects for both clinical research and AI development [8].

In addition to enhancing training data, GANs also have promising applications in medical education. A recent study showed that StyleGAN3 can generate realistic amelanotic melanoma images with 92% perceptual realism, offering a valuable tool for teaching complex diagnostic scenarios to dermatology trainees [5].

This review aims to provide a comprehensive overview of the current and emerging uses of GANs in dermatology, discussing their benefits, limitations, and future perspectives. Despite the existence of multiple reviews on artificial intelligence in dermatology, few have specifically focused on GANs and their unique clinical implications. Our contribution lies in offering a dermatologist’s perspective that bridges technical metrics with potential clinical applications, which we consider highly relevant given the current absence of clinical integration.

## 2. Materials and Methods

### 2.1. Objectives of the Work

The main objective of this work is to analyze the current state of the art regarding the use of GANs in dermatological image processing and to evaluate their impact on the diagnosis and management of skin diseases. To achieve this goal, the following specific objectives were defined:To review fundamental concepts of AI in dermatology, with a focus on the role of GANs in this field.To identify current applications of GANs for the generation, augmentation, and enhancement of dermatological images.To analyze the technical, ethical, and regulatory challenges associated with the implementation of GANs in dermatological practice and research.To explore the future perspectives of GANs and their potential for clinical integration.

### 2.2. Bibliographic Search Methodology

For this literature review, we consulted several scientific databases, including PubMed (National Library of Medicine, Bethesda, MD, USA), Web of Science (Clarivate Analytics, Philadelphia, PA, USA), Medline (U.S. National Library of Medicine, Bethesda, MD, USA), and Google Scholar (Google LLC, Mountain View, CA, USA), using the TOPA portal (Osakidetza Thesaurus of Words and Articles) of the Basque Health Service (Osakidetza, Vitoria-Gasteiz, Spain). The search strategy combined keywords such as: “GANs in dermatology,” “deep learning in dermatology,” “synthetic images in dermatology,” “medical image augmentation,” and “adversarial networks skin diseases.”

Inclusion criteria:Articles published in indexed journals or academic works (e.g., Final Degree Project (FDP) or Master’s Thesis (MT)) focused on AI and dermatology, prioritizing review articles.Publications from the last five years (2020–2025) to ensure relevance and currency, although some foundational studies published earlier were also included.Full-text availability.Articles in English or Spanish.

The PRISMA (Preferred Reporting Items for Systematic Reviews and Meta-Analyses) methodology was followed, encompassing four phases:Identification: A preliminary search was conducted in selected databases.Screening: Duplicates were removed, and titles/abstracts were reviewed for relevance.Eligibility: Full texts were evaluated according to predefined inclusion and exclusion criteria.Inclusion: Only studies that met all criteria were included, ensuring methodological rigor and thematic relevance.

This PRISMA diagram (Figure 1) summarizes the study selection process for the literature review on the use of GANs in dermatology.

For better organization and analysis of the articles included in the review, a classification was made according to the type of study (Table 1). This categorization makes it possible to distinguish the purpose, methodological approach, and level of applicability of each work, facilitating comparison between studies and the identification of trends, gaps, and needs in research on the use of GANs in dermatology.

The classification used was inspired by standard methodological criteria used in systematic reviews and adapted to the context of artificial intelligence in health.

### 2.3. Methodological Quality Assessment of Included Articles

Given the central focus of this review on GAN applications in dermatology, we not only compiled relevant studies but also evaluated their methodological rigor and scientific contribution. This qualitative analysis aimed to identify research strengths and limitations, minimize bias, and support evidence-based conclusions.

Each selected study specifically involving GANs was evaluated based on the following:Type of study (experimental, comparative, methodological, review, etc.).Objective and methodology (clarity of aim, GAN architecture, dataset characteristics, and validation strategy).Level of evidence, adapted from AI-in-health classification systems:
○Level I–II: Studies with clinical validation or comparative trials with standard reference.○Level III: Methodological studies using retrospective data or public datasets.○Level IV–V: Foundational theoretical studies or narrative reviews without experimental validation.Strengths (e.g., innovation, external validation, reproducibility, applicability).Limitations (e.g., small sample size, lack of clinical testing, limited generalizability).

This structured evaluation enabled a robust comparison between studies, highlighting trends, unmet needs, and directions for future research—particularly regarding data diversity, clinical validation, and model generalizability.

The results of this assessment are summarized in Table 2.

Although the main objective of this review is to analyze the use of GANs in dermatology, it was decided to include some studies that do not focus exclusively on this medical specialty. This decision was based on the fact that certain articles, although applied to other areas of medicine or even of a theoretical–methodological nature, provide key knowledge for the understanding, development, or evaluation of NGS models subsequently used in dermatology.

The inclusion of these papers broadens the perspective of the review and allows us to enrich the analysis of the applications and limitations of GANs in dermatology, contextualizing them within the broader framework of medical AI.

## 3. Results

### 3.1. AI in Dermatology

#### 3.1.1. Basic Concepts of AI in Medical Imaging

AI applied to medical image analysis has undergone a remarkable development in recent years, especially through advanced deep learning techniques [12,15]. There are two main types of neural networks used in AI: traditional neural networks (ANNs) and convolutional neural networks (CNNs), which have fundamental differences in their architecture, operation, and applications [2,5].

ANNs are fully connected, meaning each neuron in one layer is linked to every neuron in the next, resulting in a large number of parameters. In contrast, CNNs use local connectivity, where neurons are connected only to specific regions of the input, reducing the number of parameters and enabling efficient feature detection.

When it comes to data processing, ANNs handle input as a flat vector, which may ignore the spatial structure of data such as images. On the other hand, CNNs process structured data in two or three dimensions, allowing them to recognize spatial patterns like shapes and textures. This makes CNNs particularly effective at automatically extracting hierarchical features through layers of filters and kernels.

CNNs are more efficient in terms of parameter usage, as they reuse weights across the input space. ANNs, due to their dense connectivity, tend to generate many more parameters, which can increase the risk of overfitting and require more training data. Consequently, ANNs are often used in tasks such as classification and prediction of tabular or sequential data, while CNNs excel in visual recognition tasks like image classification, object detection, and medical image analysis.

Another key difference lies in spatial invariance. While ANNs do not inherently account for spatial relationships, CNNs are partially invariant to the position of objects in the input due to operations like pooling. Lastly, CNNs typically require more computational resources than ANNs, especially during training, as they involve multiple operations and deeper architectures. They often rely on GPUs to be trained efficiently.

Within medical imaging, CNNs have become the most widely used and effective method for automatic image interpretation due to their ability to recognize complex visual patterns. CNNs are computational models inspired by the functioning of the human visual system [2,12,40]. A CNN is a type of neural network specifically designed to process data with a mesh structure, such as images, videos, or audio signals. These networks work by successive layers performing mathematical operations called convolutions, applying specific filters on each pixel of an image to extract distinctive patterns such as edges, shapes, and textures. They are essential in computer vision tasks such as image classification, detection, and segmentation.

CNNs are mainly composed [2,12,40] of the following key elements (Figure 2):-**Convolutional layers:** This is the core of the CNN and where the main operations are performed. Each layer applies filters or kernels that apply mathematical filters on the original image to generate activation maps that highlight specific features (such as edges, textures, anomalies, or simple shapes) relevant for classification. The output of this operation generates feature maps representing the regions activated by the applied filter.-**Pooling layers**: They reduce the spatial dimensions of the processed images, keeping only the most relevant information and reducing the computational complexity. The most common techniques are max pooling (selection of the maximum value within a region) and average pooling (averaging of values).-**Fully connected layers**: After multiple convolutional and clustering layers, these layers receive the previously generated activation maps and use them to make final predictions or classifications. These layers work by integrating all previously extracted features to generate a specific diagnosis or classification.

The main advantages [2] of using CNNs are as follows:-**Automation in feature extraction**: CNNs eliminate the need to manually design filters to identify patterns in images.-**Spatial invariance**: They are able to recognize objects regardless of their position or orientation within an image.-**Hierarchy in learning**: They learn representations from simple features (edges) to complex (whole objects).

CCNs have been applied to issues such as face classification and recognition, semantic detection and segmentation in medical images [38], autonomous driving through visual analysis, and real-time object recognition. However, their use presents several challenges such as the **intensive use of computational resources** (i.e., training CNNs requires specialized hardware, such as GPUs) and the need for **large, labeled datasets** (i.e., for effective training, large volumes of representative data are required).

In dermatology, these networks have been successfully employed to automatically detect skin diseases from images obtained by digital photography or dermoscopy [12]. An emblematic example is the model developed by Google Health [20] that used CNNs trained with an extensive set of over 130,000 dermatological images to classify with high accuracy 26 different skin pathologies. This model achieved a diagnostic accuracy comparable to that of the obtained by certified dermatologists, thus demonstrating the real potential of these technological tools to support and complement clinical decision-making [12,40]. In the field of skin pathology, these models can identify pigmented and non-pigmented lesions with similar or even greater accuracy than that achieved by experienced dermatopathologists in certain specific tasks, such as differentiating squamous cell carcinomas from benign seborrheic keratoses or malignant melanomas from benign nevi [40]. In addition, due to their speed and low operating cost, CNNs facilitate rapid and reliable diagnoses, which is especially valuable in clinical settings with limited resources or high demand for care [2]. In short, the successful application of CNNs in dermatology clearly demonstrates their potential as complementary tools capable of optimizing clinical diagnosis through fast, accurate and accessible automated analyses. However, it is also essential to consider the challenges associated with their practical implementation, such as unrealistic expectations on the part of healthcare personnel or patients, ethical issues related to privacy and security of clinical data, and medico-legal aspects that must be carefully addressed before their widespread adoption [9].

#### 3.1.2. AI Applications in Dermatology

##### International Skin Imaging Collaboration (ISIC—*New York, NY, USA*)

ISIC (https://www.isic-archive.com/ accessed on 12 May 2025) is a global initiative that brings together academic institutions and industry to improve skin cancer diagnosis through the use of digital skin imaging technologies. Its purpose is to reduce skin cancer mortality by promoting standards in dermatologic imaging and fostering collaboration between the dermatology and computer vision communities.

Its main activities and resources are as follows:-**Public Image Archive**: ISIC maintains an open-access public archive containing tens of thousands of dermatologic images. This resource is invaluable for teaching, research, and the development of artificial intelligence algorithms for skin disease diagnosis and is used in most AI projects in dermatological imaging.-**Standards Development**: The collaboration works on the creation and promotion of standards for imaging in dermatology, addressing aspects such as technology, technique, terminology, privacy, and the interoperability of dermatological imaging.-**Machine Learning Challenges (ISIC Challenges)**: ISIC organizes annual competitions that invite artificial intelligence researchers to develop and evaluate algorithms for the analysis of skin lesion images. These challenges have contributed significantly to the advancement of computer-aided diagnostic techniques in dermatology, and some of them have included the use of GANs.

In addition, ISIC collaborates with platforms such as **dermoscopedia.org**, integrating images and educational resources to facilitate learning and dissemination of knowledge in dermoscopy (Figure 3).

##### AI in Dermatology

AI has been implemented in dermatology for a variety of functions, including the following:-**AI-assisted diagnosis**: Models such as the one developed by Han et al. [44] have demonstrated greater than 90% accuracy in detecting melanomas from dermoscopy images.-**Detection of** basal cell and squamous cell **carcinomas**: Here, again, the algorithm of Han et al. [44] showed an accuracy of 94.8% in the identification of malignant skin tumors.-**Differential diagnosis** in inflammatory diseases: Several models [3,5,45] allow for distinguishing between psoriasis, dermatitis, and rosacea from clinical photographs.-**Segmentation of dermatological images**: Some of the “ISIC Challenge” have driven accurate segmentation algorithms to delineate pigmented lesions and tumor margins in dermoscopic images and improve the quality of dermatological images [5].-**Disease severity assessment**: Applications such as SkinDeep-PASI use neural networks to automatically calculate the PASI index in patients with psoriasis [5]. In hidradenitis suppurativa, image-based AI has been explored to assess the Hurley stage and extent of involvement [3].-**Follow-up of disease progression**: Longitudinal follow-up tools, such as DermTrainer^®^ (Silverchair Information Systems, Charlottesville, VA, USA), allow us to assess the evolution of skin lesions by comparing serial images, which is very useful, e.g., to evaluate the efficacy of treatments (https://www.silverchair.com/news/silverchair-launches-dermtrainer/ accessed on 12 May 2025)-**AI-assisted teledermatology**: AI-enabled mobile apps allow patients to obtain a pre-assessment of skin lesions without the need for an in-person consultation. The SkinVision^®^ app (SkinVision BV, Amsterdam, The Netherlands) analyzes images of moles and suspicious lesions using AI and provides a risk recommendation to the patient. The First Derm^®^ platform (iDoc24 AB, Gothenburg, Sweden) uses AI algorithms to assist in the triage of patient-referred dermatology cases (https://www.firstderm.com/ai-dermatology/ accessed on 12 May 2025)-**Medical education and training**: Simulators such as VisualDx’s Virtual Patient^®^ (VisualDx, Rochester, NY, USA) use AI to generate clinical cases and evaluate students’ diagnostic reasoning.-**Dermatological research**: Analysis of large volumes of images from the HAM10000 project (Medical University of Vienna, Vienna, Austria) has advanced the study of patterns in common pigmented lesions [46].-**Cosmetic applications and esthetic dermatology**: Tools such as YouCam@ (Perfect Corp., Taipei, Taiwan) or Modiface@ (Modiface Inc., Toronto, ON, Canada) use AI to simulate cosmetic treatments, analyze skin aging, and predict esthetic results.

#### 3.1.3. Current Challenges and Limitations

-
**Biases in AI Models**


One of the main problems in the implementation of AI in dermatology is bias in the quality and representativeness of training data: e.g., racial or phototype bias due to lack of diversity in databases or imbalances in the representation of rare diseases. Most datasets are predominantly composed of images of light-skinned patients, which reduces the effectiveness of the model in patients with darker skin [47]. This can lead to diagnostic errors and reduced equity of care.

-
**Model Interpretability**


Deep neural networks, including CNNs used in dermatology, are often considered “black boxes,” making it difficult to explain their decisions. Recent research has explored the use of heat maps and interpretability algorithms such as Grad-CAM to improve model transparency [48].

-
**Regulatory and Legal Considerations**


The use of AI in dermatology poses regulatory challenges, especially with regard to the approval of these systems by agencies such as the FDA and EMA. It is critical to ensure that models are rigorously evaluated clinically before implementation [49]. Another added problem is the slow approval by regulatory agencies.

-
**Integration with Clinical Practice**


In this respect, two factors are essential:There is a need for interoperability with electronic medical record systems.There is a reluctance on the part of professionals to rely on automated models. In addition, clinical validation is needed, which is difficult to achieve due to the lack of multicenter studies with prospective evaluation (i.e., it would be essential to have clinical trials that demonstrate real benefit in daily practice).

In conclusion, AI has proven to be a promising tool in dermatology, with applications ranging from automated diagnosis to clinical research. However, its widespread adoption requires overcoming significant technical, ethical, and regulatory challenges. The development of robust, explainable, clinically validated and data-driven models will be key to ensuring responsible and effective integration into the dermatologic care of the future. And this is where GANs can play a very important role.

### 3.2. Generative Adversarial Networks (GANs)

#### 3.2.1. Principles and Functions of GANs

GANs were introduced by Goodfellow et al. [6] and have proven to be one of the most innovative techniques in the field of DL. These networks are based on the competition between two neural models: a generator and a discriminator.

-Generator: transforms random noise vectors (z) into synthetic images. In dermatology, it is optimized to replicate skin textures (e.g., pores or vascularization) using deep convolutional layers [50].-Discriminator: evaluates whether an image is real or artificially generated, using adversarial loss functions.

During training, the generator improves its performance to fool the discriminator, while the discriminator is refined to detect false images. This process of adversarial feedback means that over time, the synthetic images become virtually indistinguishable from the real ones (Figure 4).

#### 3.2.2. Relevant GAN Architectures in Medical Imaging

There are several GANs architectures used in medical imaging, including the following:-DCGAN (Deep Convolutional GANs): Introduces deep convolutions to improve the quality of the generated images [51]. For example, it standardizes stable training with batch normalization, generating 1024 *×* 1024-pixel images useful for analyzing melanoma patterns [8,39]. It is often used to synthesize dermoscopic patches that mimic lesion texture, helping to balance datasets and improve classifier training.-Pix2Pix: Used for image translation tasks, such as the conversion of grayscale images to color in dermatology [52]. Useful for paired image-to-image translation (e.g., clinical versus dermoscopic views when paired data exist), supporting reconstruction and education.-CycleGAN: Allows for the conversion of images between domains without the need for image pairs, useful in the synthesis of dermatoscopic dermatological images from clinical images [53]. Enables unpaired domain transfer (e.g., standardizing images from different cameras/centers), which can reduce device-related variability before analysis. The study by Han et al. [44] achieved an SSIM (Structural Similarity Index) of 0.89 in this task.-StyleGAN: Introduces fine control over the style and visual characteristics of the generated images [28,41]. Generates high-resolution, controllable images to synthesize rare phenotypes and darker phototypes for education and data balancing.

#### 3.2.3. Evolution and Improvements in Models GANs

Recent advances in GANs have revolutionized their application in dermatology and medical diagnostics, with key technical improvements in five main areas:**Training stability**

The introduction of Wasserstein GAN (WGAN) [26] with a gradient penalty solved problems of modal collapse (the model it stuck generating only one type of image although it should make many different ones) and instability (training a GAN is like a tug-of-war between the generator and the discriminator; if one becomes too strong, the other remains unlearned) in conventional GANs. This approach allows for the training of complex architectures while maintaining continuity in the loss surface—critical for medical applications where consistency is vital [54].

2.
**Image quality and resolution**


Models such as Progressive Growing GAN (PGGAN) generate high-resolution images (512 × 512 pixels), which have been used to preserve critical anatomical details in medical imaging [41]. StyleGAN2, on the other hand, has demonstrated a superior ability to produce very-high-resolution (1024 × 1024 pixels) dermatoscopic images, capturing subtle patterns of melanomas with clinical fidelity [55].

3.
**Semantic control**


Semantic control, facilitated by techniques such as attention maps and latent encoders, allows for the relevant features of medical images to be selectively modified. For example, these techniques can be used to adjust morphological parameters in dermatological lesions or to preserve critical anatomical structures during translation between modalities [56].

4.
**Conditionality**


cGANs have demonstrated efficacy in tasks such as multimodality translation (e.g., conversion of T1w to FLAIR images in MRI) and generation conditional on specific diagnoses. These capabilities enable the generation of synthetic images that simulate specific clinical conditions and facilitate the training of medical classification models [42,52].

5.
**Multimodality**


Hybrid systems [23] that combine different sources of information, such as clinical images and histopathological data, make it possible to generate composite or correlated representations. These techniques have been applied to combine multiple MRI sequences or to improve spatial resolution in angiographic images [57].

In dermatology, StyleGAN2 has enabled the generation of large synthetic datasets, such as more than 26,000 synthetic dermatoscopic images used to train AI classifiers. This has improved diagnostic sensitivity by 15% over conventional datasets and has facilitated the creation of balanced image banks that include rare pathological variants [46].

#### 3.2.4. Limitations and Challenges in Implementing GANs

GANs have demonstrated enormous potential in the generation of synthetic medical images, but their implementation faces several technical, ethical, and operational challenges. The main limitations and challenges are detailed below:-**Lack of explainability and interpretability**: The “black box” of generative models makes their clinical validation difficult [3,9,46]. GANs present difficulties in explaining how they generate specific synthetic images. In the medical context, this lack of interpretability can be a major obstacle to their clinical adoption, as practitioners need to understand how the images are produced in order to rely on them as diagnostic tools.-**Possibility of bias in the generated data**: If the training dataset is limited or not representative, GANs can amplify existing biases [47]. Also in medical imaging, data can vary significantly depending on the quality of the equipment used, the experience of the technician generating the data, and the presence of artifacts. These inconsistencies can “foul” the data and compromise the accuracy of trained models [18,30].-**Regulation and clinical approval**: The lack of specific regulatory standards for generative models poses obstacles to their clinical implementation. Moreover, although these images do not correspond to real patients, their misuse could raise privacy and security issues. In addition, there is a risk of GANs being used to create false or misleading medical content [35].-**Costs and operational complexity**: The creation and maintenance of standardized databases to train GANs is a resource-intensive process. This includes technical harmonization between different sources, accurate annotation, and compliance with legal and ethical regulations. These difficulties limit the availability and representativeness of the data needed to develop reliable tools [9,41].

### 3.3. GANs in Dermatology

GANs have been used in dermatology for various applications [8,43], such as synthetic imaging of skin lesions, improved image segmentation, and color standardization in dermatological images.

The main algorithms and GANs investigated in Dermatology are as follows:DermoCC-GAN [29]:

Application: Standardization of color in dermatological images.

Characteristics: Combines a customized heuristic algorithm with an antagonistic generative network to correct for variability in image illumination conditions. This generative network improves color constancy in dermatological images to standardize illumination by aiding lesion classification and segmentation and outperforming traditional color normalization algorithms (Figure 5).

2.Efficient-GAN (EGAN) [24]

Application: Skin lesion segmentation.

Characteristics: It uses an adversarial learning architecture with a generator that incorporates a composite scaling path with excitation attention and asymmetric lateral connections. It includes a morphology-based loss function to generate smoother lesion contours. It is notable for its accuracy and a lightweight version called Mobile-GAN, suitable for devices with low computational power (Figure 6).

3.DermGAN [22]

Application: Generation of synthetic clinical images of skin with specific pathologies.

Characteristics: Based on the Pix2Pix architecture (an image translation technique designed to transform images from one domain to another by supervised learning), this application allows for variance in the skin color, size, and location of the condition in the synthetic images, while maintaining the fidelity of the pathology represented (Figure 7).

4.Skin Lesion Style-Based GANs [17]

Application: Skin lesion image synthesis for improved classification.

Characteristics: Modify the style control structure and noise input in the generator to produce high quality images, improving the performance of skin lesion classification models. The generated images were visually realistic and useful for augmenting the training data. The classifier trained with real and synthetic images obtained higher accuracy than the one trained only with real images (Figure 8).

5.GLSFA-GAN (Generative Adversarial Network with Global and Local Semantic Feature Awareness) [34].

Application: Skin lesion segmentation.

Characteristics: It incorporates modules to fuse local features from multiple scales and extract the global context, improving accuracy in the segmentation of lesion areas (Figure 9).

6.GAN with Hybrid Loss Function [23].

Application: Data enhancement for dermoscopic image classification.

Characteristics: They introduce a hybrid loss function to improve the generation of synthetic images, evaluating its effectiveness in the classification of seven types of skin lesions. The authors propose an unsupervised data augmentation solution that uses GANs and associated techniques in their latent space to generate controlled semantic variations in dermatoscopic images that are used to augment the training dataset. Recently, Veeramani et al. [33] proposed something similar—a new GAN called MELIIGAN—to improve the quality of dermoscopic images and facilitate fine feature extraction. This approach uses stacked residual blocks to handle large scaling factors and reconstruct intricate details. In addition, it incorporates a hybrid function of loss with total variation regularization and the Charbonnier function, which is more robust than the mean square error (Figure 10).

7.“Conditional GANs for Dermatological Imaging [19].

Application: Generation of images of skin lesions conditioned by segmentation masks.

Characteristics: A **cGAN** is the same as a GAN, but with an additional condition. That is, you tell it what you want it to generate. It incorporates segmentation masks in the generation process to improve the structure of the lesions in the synthetic images (Figure 11).

There are many studies [10,11,13,14,21,32] using these techniques that have focused on the diagnosis of skin cancer, especially in the detection of melanomas.

### 3.4. Practical Applications in Clinical Dermatology

Once the latest advances in AI and GANs in dermatology are known, as clinical dermatologists, we ask ourselves “how can all this knowledge help us in our actual clinical practice?” All of the following come to mind:-Clinical support in consultation or automated triage: We could augment datasets and generate realistic synthetic images of skin lesions (melanoma, psoriasis, acne, etc.) to improve the training of AI algorithms when there is little data [16]. This will improve the accuracy of diagnostic models.-Simulation of disease progression [19]: We could show how a lesion would evolve over time if left untreated (melanoma, ulcers, psoriasis, etc.). This would be an extraordinary support in clinical decision-making, in visual explanations for patients, and in the selection of more aggressive treatments if unfavorable evolution is predicted.-Reconstruction of poor-quality images: We could improve blurred, pixelated, or poorly illuminated photos (e.g., sent by patients). This would allow us to make better remote diagnoses (teledermatology), even with suboptimal images.-Standardization of color and image conditions [30]: We could automatically correct for differences in light and skin tone or contrasts between images taken under different conditions. This would facilitate comparisons between visits, evolutionary follow-up, and improve the accuracy of AI or our own clinical assessment.-Lesion segmentation: We could automatically separate the lesion from the background (healthy skin) in an image. This could lead to a more objective measurement of the affected area (ulcer, psoriasis, eczema…), follow-ups on the response to treatment, and more accurate clinical documentation.-AI-assisted classification [25,27]: We could generate images to train systems that then classify lesions (melanoma vs. nevus [25], types of acne, etc.). This could be a source of diagnostic support in consultation, which is especially useful in patients with multiple lesions, or in the triaging of suspicious cases.-Medical education and training: We could create synthetic didactic images of rare cases or cases at different stages. These could serve as educational tool for students, residents, or even other specialists (e.g., primary care physicians).-Improve balance in clinical datasets: We could generate images of under-represented populations (e.g., dark skin tones [37]). This could contribute to more inclusive AI systems and avoid bias in automated diagnostics.-Creation of interactive tools for patients: We could show the patient how their skin will look with certain treatments or if left untreated (e.g., progression of rosacea, acne, vitiligo). This could improve adherence to treatment programs and understanding of the disease, especially in digital natives or young patients.-Simulation of esthetic or surgical results: We could generate simulated images of how the skin will look after an intervention (laser, cryotherapy, surgical resection). This could lead to clearer communication with the patient, the management of expectations, and support for informed consent.-Development of AI clinical applications: We could use generated images to create or enhance automatic assessment apps (e.g., for acne patients who send weekly photos). This could assist in the remote monitoring of chronic patients and the optimization of office resources and lead to more accurate referrals.

In addition to academic research, several commercial initiatives are already incorporating GAN-based solutions into dermatological tools. These developments often align with the use cases described in previous sections—such as data augmentation, rare lesion synthesis, or image standardization—and are designed to support clinicians in diagnostic, educational, and operational workflows.

Table 3 summarizes selected commercial applications identified through open-source and market searches (Google search/official websites), grouped according to the eleven use cases discussed. This overview illustrates the real-world translation of GAN technology into products aimed at clinical dermatologists and highlights the increasing relevance of generative models in industry-led innovation.

### 3.5. Reliability Assessment and Ethical Challenges

#### 3.5.1. Validation of Images Generated by GANs

Synthetic images generated by GANs must undergo rigorous validation processes [8] before being used in clinical settings. There are multiple strategies to assess their quality (Table 4).

##### Comparison with Real Images Using Metrics:

In this review, some metrics help us judge how “real” synthetic images look compared with real ones, others reflect structural fidelity in paired tasks like denoising/super-resolution or estimate realism/diversity. These metrics are proxies: they should be interpreted alongside task-specific, clinically meaningful outcomes. The following are commonly used:-Fréchet Inception Distance (FID) [23,58]: FID measures the degree of similarity between images generated by a GAN and real images by comparing not only pixels but also deep features extracted by a trained neural network (usually Inception v3). The real images and the images generated by the Inception network are passed; this network extracts “features” (internal representations) from each set of images and compares the statistical distributions of those features (mean and covariance) using a mathematical formula called Fréchet distance. The lower the FID, the better; it indicates that the generated images are more similar to the real ones. An FID of 0 would be ideal (i.e., the fake images are indistinguishable from the real ones).-Structural Similarity Index (SSIM) [19,33]: The SSIM compares two images and evaluates their similarity structurally. It is very useful, for example, to see if a generated image is visually similar to an original image (e.g., in super-resolution or denoising tasks). It analyzes three things between the two images: luminance, contrast, and structure. A value between 0 and 1 is calculated, where 1 means identical images, and values close to 0 are completely different. A value greater than 0.9 is usually already considered to represent a high structural similarity.-Quality and realism metrics:
Inception Score (IS): Uses a pre-trained Inception network to calculate the entropy of the predicted labels on the generated images, but its limitation is that it does not compare directly with real images [26,58,59].Peak Signal-to-Noise Ratio (PSNR) [19,28,33]: Measures the quality of the reconstruction (how much the generated image resembles the original, pixel by pixel), but it does not capture perceptual differences well; it only measures absolute intensity differences.Mean Squared Error (MSE): Measures the average squared difference per pixel between real and generated images, but like PSNR, it does not reflect perceptual quality [19,28,33]-Perceptual metrics:
Learned Perceptual Image Patch Similarity (LPIPS) [26,34,48]: Measures perceptual similarity between images by calculating the distance between activations of deep neural networks (such as AlexNet, VGG) between two images. It has the advantage of a better correlation with human perception than SSIM or PSNR.Perceptual Path Length (PPL) [34,50,55]: Measures the smoothness of the latent space of the GAN (how the image changes when interpolating between two latent vectors) and is used to evaluate the quality of the latent space in GANs such as StyleGAN.-Diversity Metrics:
Precision and Recall for Distributions [26,43,50]: Measures how realistic the generated images are and the diversity of the generated sample with respect to the real ones. It helps to evaluate the balance between fidelity and diversity, something that FID does not distinguish well.Mode Score [26]: An extension of the Inception Score that penalizes the lack of class coverage and is used when you want to evaluate class diversity (e.g., digits 0 to 9 in MNIST).-Specific metrics in medical imaging:

The Dice Coefficient and Jaccard Index [22,24,34] measure the overlap between images (useful in segmented images or in controlled generation) and are used in GANs to compare synthetic images with real anatomical structures.

##### Evaluations by Clinical Experts

The use of synthetic images generated by GANs in dermatology requires rigorous validation by clinical experts. In this context, dermatologists review the generated images to determine whether they are indistinguishable from the real ones, which is a critical step to ensure the quality and diagnostic utility of these images. Tschandl et al. [46] conducted a study using the HAM10000 dataset, which is composed of dermoscopic images of various skin lesions. The experts evaluated both real and synthetic images generated by Convolutional Neural Networks (CNNs) and GANs, demonstrating that well-trained synthetic images can be almost indistinguishable from real ones. However, this process is highly dependent on the clinician’s experience and the quality of the generative algorithm used.

##### Testing on Models from Classification

Another key approach to evaluating the utility of synthetic images is their integration into automated classification and diagnostic models. Bissoto et al. [59] analyzed how data generated by GANs can improve the performance of classification algorithms in tasks such as melanoma detection. In their study, models were trained with datasets augmented by synthetic images, which allowed the authors to increase the diversity of the training set and to mitigate problems such as inter-class imbalance.

The results showed that models trained with augmented data achieved significant improvements in metrics such as diagnostic accuracy and sensitivity. However, limitations were also identified: the generated images may introduce biases if they do not adequately represent rare or complex clinical cases. In addition, the importance of proper preprocessing and the careful selection of metrics to assess the real impact of synthetic data on model performance was highlighted.

#### 3.5.2. Biases and Problems in Generative Models

Despite its advantages, the implementation of GANs in the medical domain presents significant challenges related to biases in the training data and problems inherent to generative models, such as the following:-**Limitation in the representation of skin phototypes**: One of the most prominent problems in the use of GANs for medical imaging is the lack of representativeness in the training data, especially in relation to skin phototypes. Most of the datasets used to train these models are biased towards lighter skin tones, which may reduce the effectiveness of the images generated for patients with darker skin. According to Adamson and Smith [47], this bias limits diagnostic and therapeutic capability in under-represented populations, perpetuating inequities in access to advanced artificial intelligence-based tools. This problem is especially critical in dermatology, where the visual characteristics of skin lesions can vary significantly depending on the patient’s skin tone. The Fitzpatrick scale [60], used to classify phototypes according to the amount of melanin and sensitivity to the sun, could serve as a benchmark for developing more diverse and representative datasets. However, its implementation requires a coordinated effort to collect clinical images that include all phototypes, from I (very light skin) to VI (very dark skin).-**Risk of overfitting to specific patterns** which may affect the variability of the generated images. Overfitting is another critical challenge in GAN training. This phenomenon occurs when the model learns specific patterns from the training dataset and loses the ability to generalize to new data. In the medical context, this can result in synthetic images that reflect only features present in the original data, reducing the variability needed to adequately represent different clinical cases [61].

For example, if a GAN is trained exclusively with images from a single type of medical equipment or specific population, the images generated may not be useful for other clinical contexts. To mitigate this problem, it is essential to implement techniques such as dropout or regularization during model training to improve the model’s generalizability [61]. Dropout addresses this problem by randomly “turning off” certain neurons during the training process, following a predefined probability.

-**Difficulties in model calibration**: Although GANs are capable of generating highly realistic images, they may contain incorrect or irrelevant clinical information. Yi et al. [8] point out that GANs may generate artifacts or anatomical inconsistencies that do not correspond to real features observed in patients. This represents a significant risk for clinical applications, as synthetic images could induce diagnostic errors if used without proper validation.

In addition, accurate model calibration is essential to ensure that the generated images are useful from a clinical point of view. This involves carefully tuning the generator and discriminator parameters to avoid problems such as *mode collapse*, where the model produces a limited variety of samples. The integration of medical experts during the validation process can help to identify these inconsistencies before the images are used in clinical applications.

Nevertheless, it is important to emphasize that these algorithmic metrics, while useful for benchmarking, do not by themselves guarantee clinical utility. To date, clinical validation of GAN-based models in dermatology remains scarce, and few studies [22,46,59] have evaluated patient-centered or clinically meaningful outcomes. Future work should therefore prioritize prospective validation trials and outcome measures that directly reflect diagnostic accuracy, safety, and usefulness in real-world dermatological practice.

#### 3.5.3. Ethics and Regulations in the Use of Synthetic Imaging

##### Ethical Considerations

The integration of GANs in dermatology raises several ethical concerns [35,36]. These include the risk of bias amplification due to the under-representation of certain skin types in training datasets, the potential misuse of GANs for deep fakes [31] or fraudulent image manipulation, and the lack of transparency inherent to black-box models. Moreover, synthetic datasets used for model training or validation may not always ensure proper informed consent or data provenance, challenging the ethical principles of autonomy and non-maleficence. The deployment of GANs in clinical contexts also requires critical evaluation of fairness and explainability to safeguard trust and accountability in AI-assisted dermatology. In addition to these general concerns, several frameworks provide more concrete orientation. In the European Union [62,63], the Artificial Intelligence Act (AI Act, 2024) explicitly classifies medical AI as “high risk” and highlights the need for human oversight, transparency, and accountability. In the United States [64], the FDA regulates AI under its Software as a Medical Device (SaMD) framework, which emphasizes good machine learning practices and continuous oversight. Recent case discussions in dermatology also underline the importance of addressing bias in under-represented skin types and ensuring proper informed consent when synthetic images are created or used for training.

##### Regulatory Landscape


**European Union—AI Act and MDR**


In the European Union, the proposed Artificial Intelligence Act (AI Act) represents a pioneering attempt to regulate AI systems based on their potential risk to fundamental rights and safety. According to the AI Act, systems used in healthcare, including those for diagnostic support or image generation, are considered high-risk and must comply with strict requirements for transparency, human oversight, robustness, and data governance [62]. In parallel, medical AI tools must also comply with the Medical Device Regulation (MDR 2017/745), which mandates performance validation, clinical evaluation, and post-market surveillance. For GANs, which often fall into gray areas of regulation due to their generative nature, their classification as Software as a Medical Device (SaMD) remains under discussion [63].


**United States—FDA Digital Health Initiatives**


In the U.S., the Food and Drug Administration (FDA) regulates AI-based medical tools under the Digital Health Software Precertification Program and the Software as a Medical Device (SaMD) framework [64]. While no GAN-based tools have yet received specific approval, the FDA has issued guidance on “Good Machine Learning Practices” (GMLP), emphasizing transparency, reproducibility, and continuous learning oversight [65]. The GANs used for medical image generation would likely be scrutinized for their intended use, i.e., for diagnostic aid or training, and evaluated based on clinical validation, traceability, and bias mitigation strategies.

##### Summary and Perspectives

Current regulatory efforts recognize the transformative potential of AI in dermatology but are still evolving in response to novel architectures like GANs. While the EU adopts a more prescriptive and precautionary approach through the AI Act and MDR, the U.S. emphasizes adaptive oversight via guidance and pre-certification programs. For dermatologists and developers, early engagement with regulators and adherence to ethical AI design principles will be critical for the responsible adoption of GANs in clinical workflows.

The following table (Table 5) provides a comparative overview of the current regulatory landscape in the European Union (EU) and the United States (U.S.), highlighting key legislation, classification approaches, AI-specific strategies, and validation requirements. This comparison is essential to understand both the opportunities and the limitations imposed by each system, and it underscores the importance of harmonizing global regulatory efforts for emerging AI technologies in dermatology.

### 3.6. Future Perspectives and Clinical Applications

#### 3.6.1. Integration into Clinical Practice

The integration of GANs in clinical dermatology represents not only a technological advance but also an organizational and ethical challenge [35,36]. Effective implementation of these tools requires overcoming several barriers:Clinical and regulatory validation: Before being incorporated into routine care, GANs-based applications must undergo rigorous clinical trials that evaluate their safety, diagnostic efficacy and added value over conventional methods. In addition, they must comply with regulatory standards, such as the European Union’s Medical Device Regulation (MDR) or FDA approval in the USA.Acceptance by healthcare personnel: For these technologies to be adopted, it is essential that dermatologists understand how they work, their limitations and their clinical applicability. Training in artificial intelligence (AI) and interpretation of results is key to fostering effective human–machine collaboration [46].Technological infrastructure and digitization: Implementation of GANs models requires a robust computing, storage, and networking infrastructure. The hospitals must have interoperable and secure systems that allow for the seamless integration of these tools [40].Interoperability with electronic health record systems (EHRs): GAN-based models can be integrated with EHRs to facilitate automated documentation, improve continuity of care, and provide more accurate assisted diagnoses [5]. This could also streamline screening for suspicious lesions through automatically generated clinical alerts.

##### Potential Impact on Assisted Diagnosis by AI

The use of GANs in dermatology has great potential to improve diagnostic accuracy and optimize healthcare resources. Some notable applications include the following:-Generation of extended and balanced datasets [59]: GANs can synthesize high-quality dermatological images with a variety of pathologies, which allows for the balancing of datasets with a scarcity of examples (e.g., images of rare diseases), reducing bias in deep learning models.-Enhanced teledermatology: The combination of GANs with telemedicine platforms allows for the simulation of diverse clinical cases, facilitating the training of models operating in remote contexts. This is especially useful for regions with limited access to dermatologists [20].-Support in differential diagnosis: GANs can generate images that represent different stages of a disease or atypical variations, helping dermatologists to recognize complex patterns and improve their clinical judgment. This could help reduce diagnostic errors in difficult cases [32].

#### 3.6.2. Future Lines of Research

As GANs evolve, new research opportunities are opening up in the dermatological field. Some future lines include the following:-Improving explainability: One of the main criticisms of AI systems is their opacity. Therefore, GAN models are being developed [48] that integrate visual interpretation mechanisms, such as attention maps or Grad-CAM techniques, which allow for the identification of which regions of the image have influenced the model’s decision.-Equity and inclusion in datasets: To prevent algorithms from perpetuating inequalities, synthetic generation strategies focused on under-represented populations, including different ethnic groups and skin phototypes, are being promoted. Diversity in training data is essential to ensure truly equitable AI [47,60].-Personalized medicine [2]: GANs could be used to predict treatment response in chronic inflammatory diseases such as psoriasis or severe acne by simulating clinical evolution with and without treatment. This would open the door to individualized therapeutic decision models.-Collaboration between dermatologists and engineers will be key to ensure clinically relevant applications.

Finally, beyond GANs, other imaging modalities, such as Hyperspectral Imaging (HSI), are emerging as promising complementary tools for dermatologic diagnostics. Although recent papers on HSI were published after the cut-off date of our search (April 2025) and were therefore not included in this systematic selection, we acknowledge their relevance. Beyond the applications of GANs alone, hyperspectral imaging (HSI) represents a promising complementary modality in dermatologic AI. HSI captures spectral signatures across dozens or even hundreds of wavelengths, enabling the detection of subtle biochemical and structural differences in skin lesions that are invisible to conventional RGB imaging. When combined with GAN-based models, these high-dimensional spectral datasets could be exploited to generate synthetic hyperspectral images, augment under-represented spectral patterns and improve the robustness of classifiers trained on multimodal inputs. Such a synergistic approach may enhance early cancer detection, standardize image variability across acquisition devices, and open the door to novel diagnostic workflows where GANs support both spectral reconstruction and realistic data augmentation. Integrating GANs with HSI therefore holds potential to advance precision dermatology by uniting morphological and biochemical insights within a single AI framework.

## 4. Conclusions

### 4.1. Summary of Findings

The present work has explored the use of GANs in dermatology, highlighting their impact on the generation of synthetic images, the improvement of AI-assisted diagnosis and the anonymization of clinical data. Multiple clinical applications have been identified, ranging from data augmentation for training AI models to prediction of skin disease progression.

Results obtained in several studies suggest that GANs can improve the diagnostic accuracy of deep learning models and optimize the quality of medical images used in dermatology. In addition, these networks present great potential in preserving patient privacy by creating synthetic images that retain essential clinical features without compromising sensitive data.

### 4.2. Limitations of the Study

Despite following a structured methodology based on the PRISMA guidelines, this review has some limitations inherent to the selected approach.

First, the time criterion established (2020–2025) allowed us to focus the search on current studies, thus ensuring the relevance of the findings. However, this restriction could have excluded key papers published in previous years, especially between 2017 and 2019, when fundamental advances in the use of GANs in dermatology occurred. Although some earlier articles were included because of their relevance, their incorporation was not performed systematically. Second, the results were limited to articles published in English and Spanish. This decision, although it responds to the linguistic competence of the reviewer, may have excluded valuable scientific literature in other languages, such as German, Chinese, or Japanese, from countries with outstanding production in the field of medical artificial intelligence.

Furthermore, the review was restricted to articles with full access to the text, which may have introduced an availability bias, especially in databases such as Google Scholar, where many references are not fully accessible. This criterion may have affected the completeness of the review by leaving out potentially relevant studies.

Another limitation lies in the preferential selection of review articles and academic papers (FDP, MT), which may have reduced the representation of some recent original studies not yet included in reviews, or in the process of publication.

On the other hand, although broad keywords were defined and MeSH terms such as “GANs in dermatology,” “deep learning in dermatology,” “synthetic images in dermatology,” “medical image augmentation,” and “adversarial networks skin diseases” were combined, terminological variability in this emerging field may have resulted in the exclusion of publications that use alternative nomenclature to describe related techniques.

### 4.3. Recommendations for Future Research

Based on the analysis performed in this review, the following recommendations are proposed to guide future lines of research in the use of GANs applied to dermatology:-Formation of **multidisciplinary teams**: It is essential to foster collaboration between dermatologists, computer engineers, biomedical engineers, statisticians, and artificial intelligence experts. The correct implementation of GANs models in clinical settings requires not only sound technical development but also a thorough understanding of the dermatologic diagnostic processes, clinical needs, and ethical issues associated with the use of synthetic medical images. The presence of multidisciplinary teams ensures a more robust experimental design, a validation more in line with clinical practice, and a proper interpretation of the results.-Increasing the **diversity and representativeness** of the data: Many current studies use public databases with an under-representation of high phototypes or rare pathologies. It is crucial to encourage the collection and sharing of diverse and well-annotated dermatologic data to improve the generalizability of models and avoid algorithmic biases.-**Clinical validation** and comparison with specialists: It is recommended that the generated models be evaluated not only by quantitative metrics (FID, IS, IoU…) but also through comparative studies with dermatologic clinical practice. This includes the participation of specialists to assess the diagnostic utility of the synthetic images or of the segmentation or classification models trained with them.-Exploration of **new clinical applications:** beyond diagnosis, GANs could have applications in medical education, image quality improvement, dataset generation for rare diseases, or pre-surgical simulations. Future studies should explore these possibilities.-Development of **accessible and explainable tools**: It is recommended to prioritize the development of models that are not only accurate but also interpretable and understandable for clinicians. The algorithmic transparency and the possibility of auditing the model’s decisions will facilitate its acceptance and use in practice.-Ethical, legal, and regulatory **considerations**: Any development should incorporate from its early stages an assessment of the ethical, legal, and privacy risks, especially in relation to the generation of synthetic images of patients. Future studies should align with emerging regulatory frameworks on AI in healthcare.

In a broader context, it is worth noting that generative models have been extensively reviewed in other areas of healthcare, such as during the COVID-19 pandemic [66]. By contrast, our review was intentionally designed to focus on dermatology, aiming to translate these highly technical advances into clinically meaningful insights. In doing so, we hope to raise awareness among dermatologists and stimulate greater engagement with a rapidly evolving field that holds considerable promise for future integration into practice.

In conclusion, GANs represent an innovative tool with the potential to transform the field of dermatology. However, their successful implementation will depend on overcoming the technical, ethical, and regulatory challenges identified in this study.

## Figures and Tables

**Figure 1 bioengineering-12-01113-f001:**
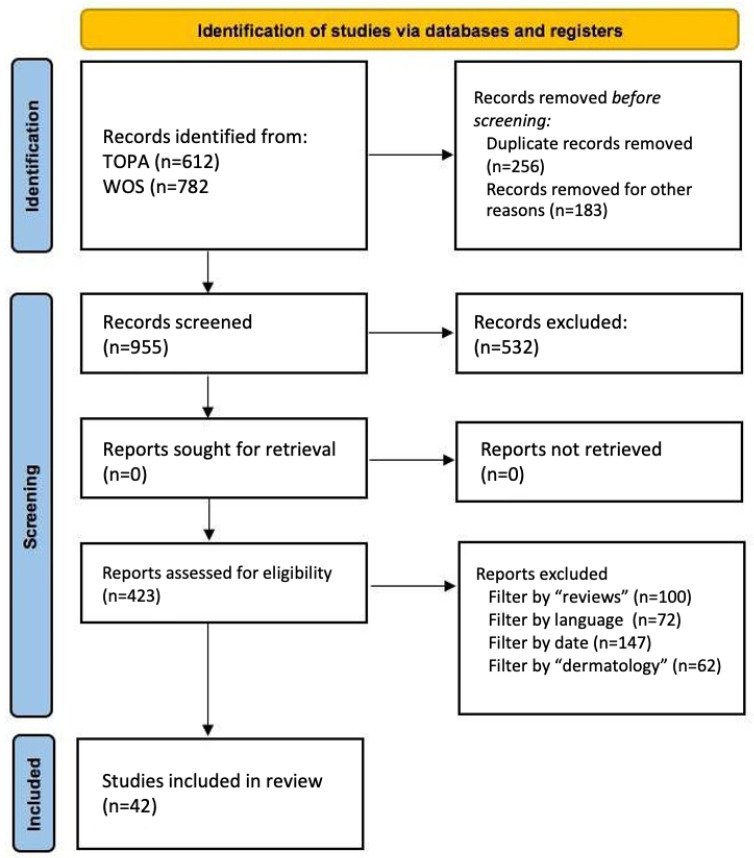
PRISMA diagram with bibliography selection process.

**Figure 2 bioengineering-12-01113-f002:**
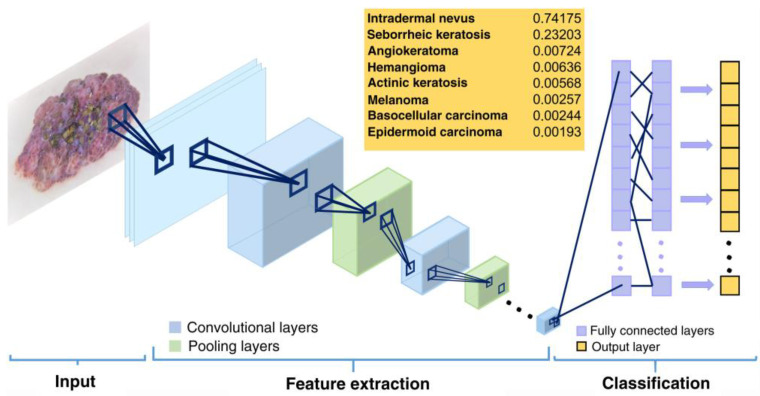
Illustration of the operational mode of a CNN [12].

**Figure 3 bioengineering-12-01113-f003:**
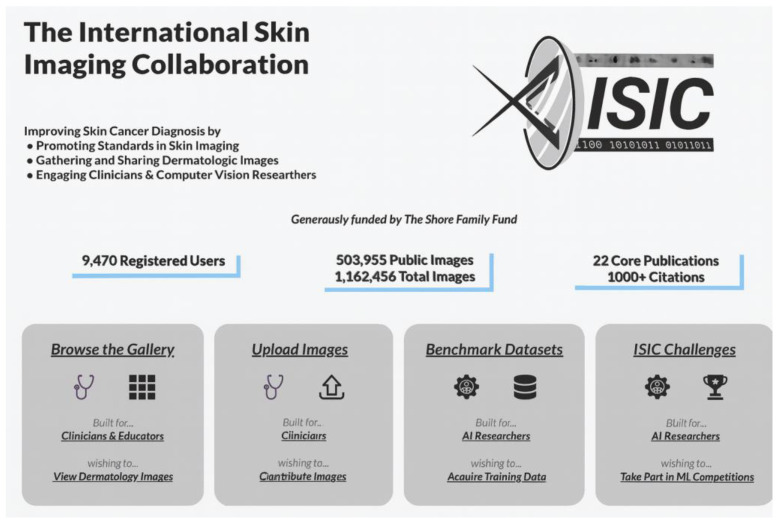
“Screenshot” of the ISIC website.

**Figure 4 bioengineering-12-01113-f004:**
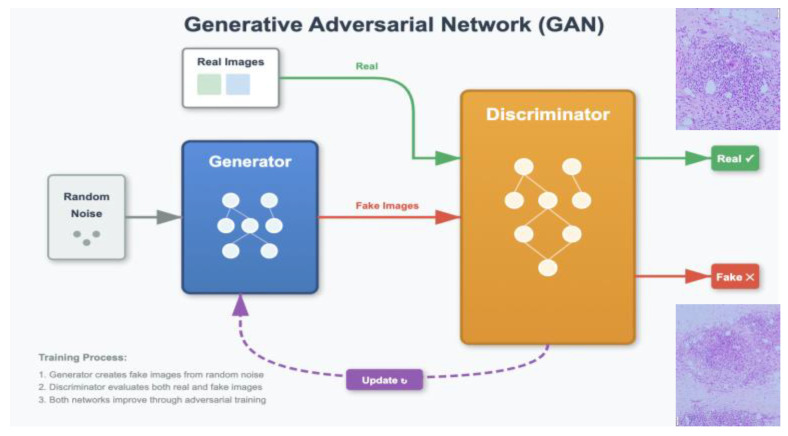
Architecture and Training Process of a Generative Adversarial Network.

**Figure 5 bioengineering-12-01113-f005:**
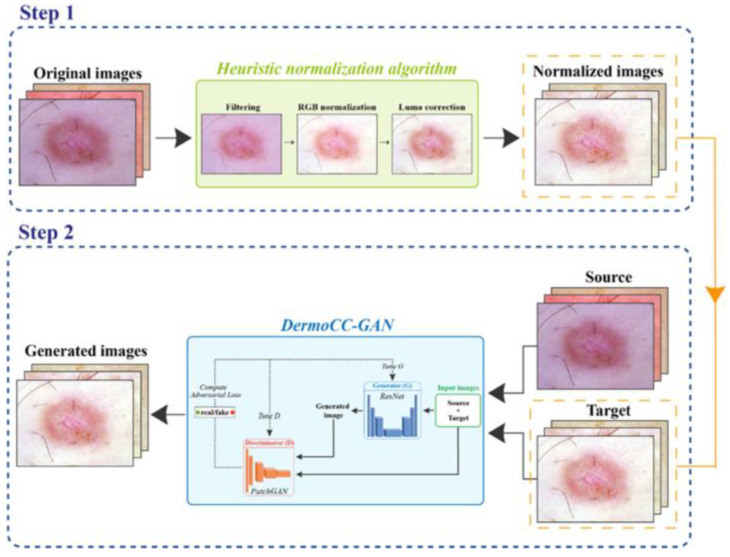
Dermatological color constancy generative adversarial network (DermoCC-GAN) [29].

**Figure 6 bioengineering-12-01113-f006:**
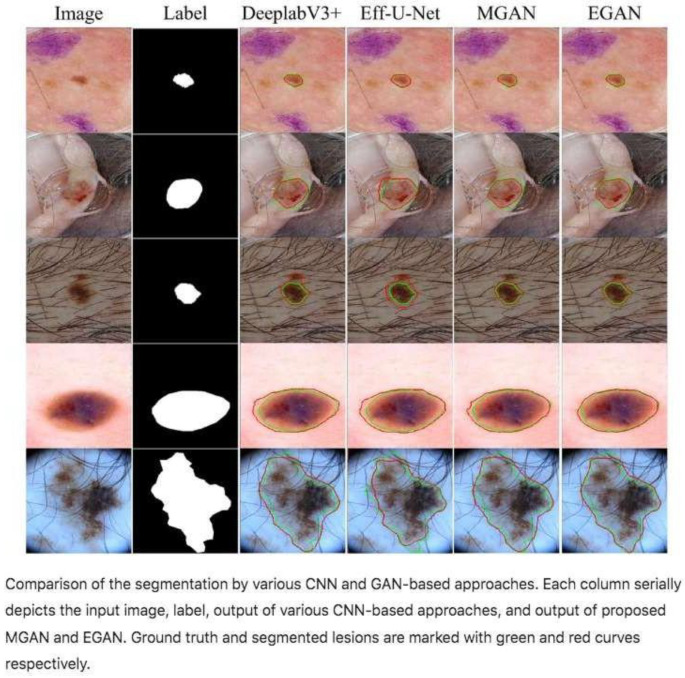
Segmentation models with the two proposed GANs: MGAN and EGAN [24].

**Figure 7 bioengineering-12-01113-f007:**
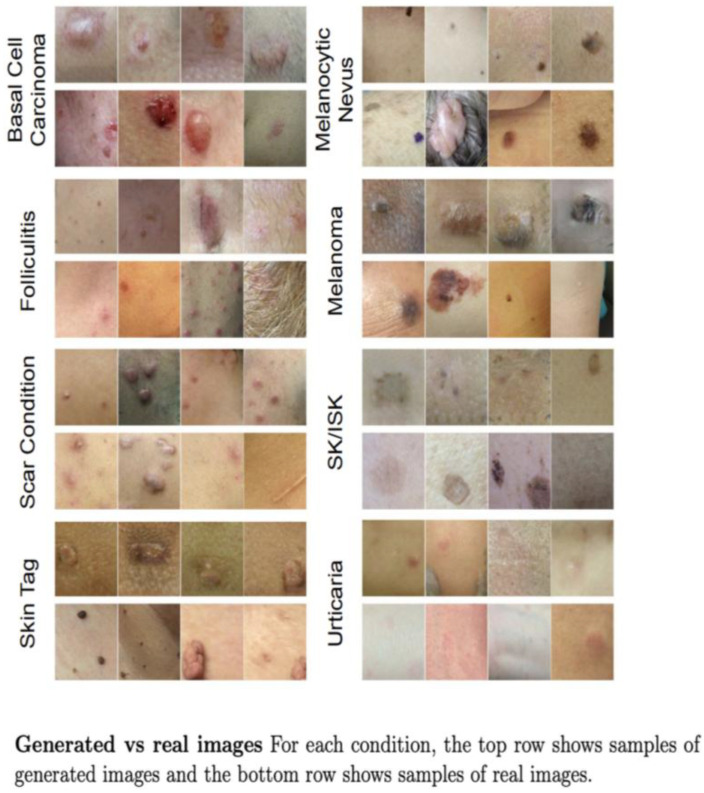
Examples of real and synthetic images with the DermGan model [22].

**Figure 8 bioengineering-12-01113-f008:**
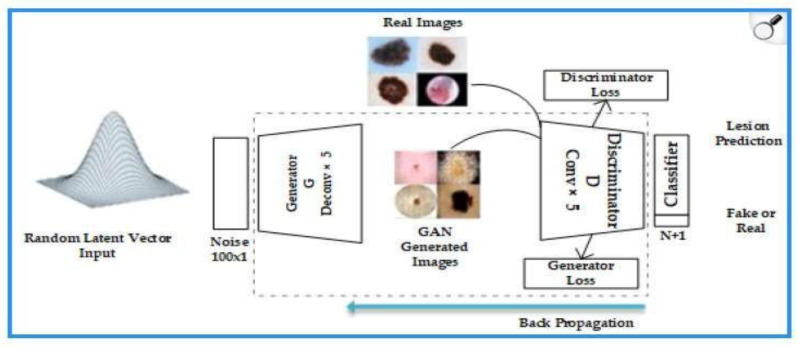
Flowchart proposed by the DCGAN architecture for skin lesion classification [17].

**Figure 9 bioengineering-12-01113-f009:**
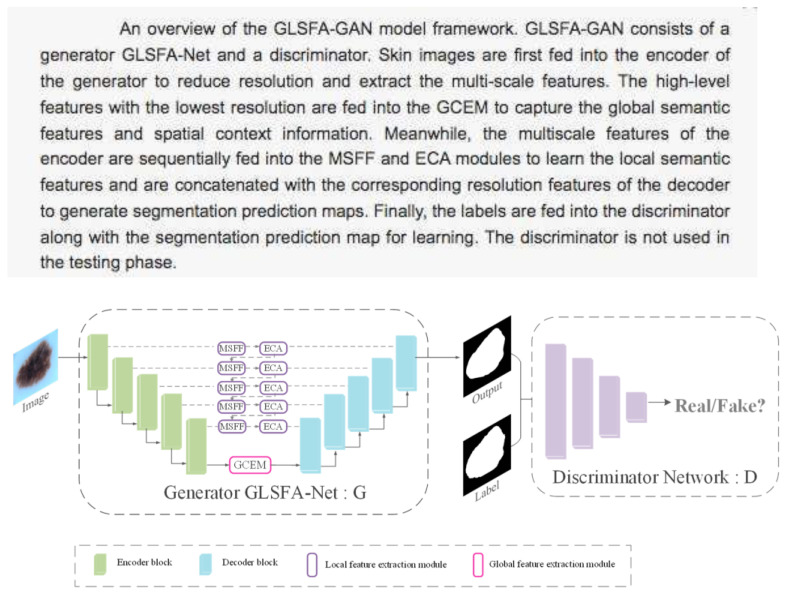
Schematic of the GLSFA-GAN model [34].

**Figure 10 bioengineering-12-01113-f010:**
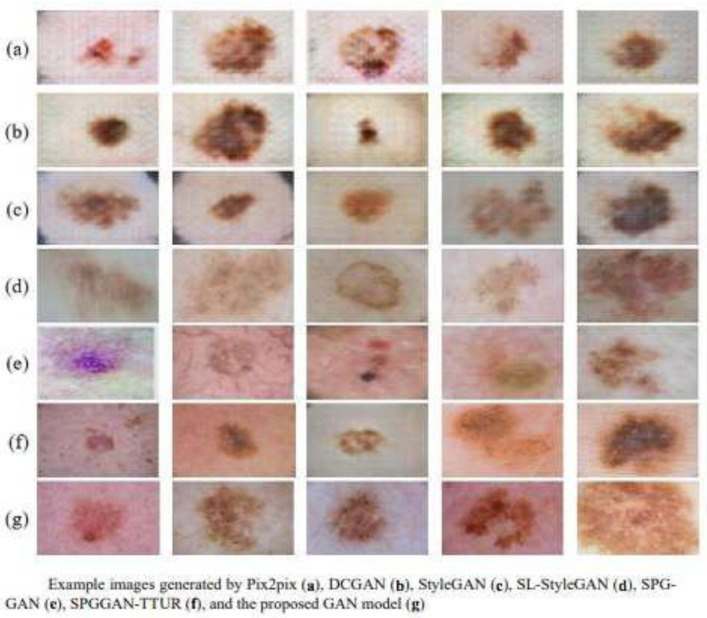
Comparison of images with different GANs and the one proposed by Goceri, E. [23].

**Figure 11 bioengineering-12-01113-f011:**
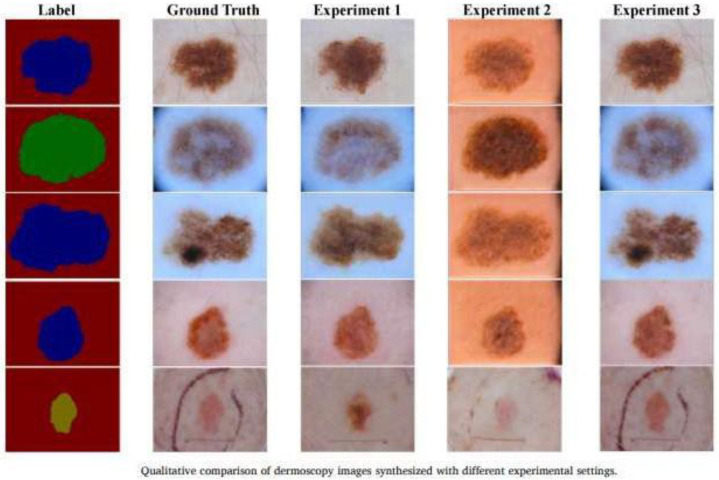
Dermatoscopic images: Qualitative comparison of the Ding. S et al. model [19].

**Table 1 bioengineering-12-01113-t001:** Selected bibliography according to type of study.

Type of Study	Number of Articles	Authors and Year
Systematic and narrative reviews	13	Behara et al. (2024) [9], Daneshjou et al. (2022) [7], Furriel et al. (2024) [10], Hermosilla et al. (2024) [11], Iglesias-Puzas and Boixeda (2020) [12], Wongvibulsin et al. (2024) [4], Kourounis et al. (2023) [2], Liu et al. (2025) [13], Li et al. (2022) [5], Nazari and Garcia (2023) [14], Omiye et al. (2023) [3], Vardasca et al. (2024) [15], Yi et al. (2019) [8]
Methodological studies (development and validation of algorithms)	20	Alshardan et al. (2025) [16], Behara et al. (2023) [17], Carrasco Limeros et al. (2022) [18], Ding et al. (2021) [19], Esteva et al. (2017) [20], Efimenko et al. (2020) [21], Ghorbani et al. (2020) [22], Goceri (2024) [23], Innani et al. (2023) [24], Heenaye-Mamode Khan et al. (2022) [25], Perez and Ventura (2023) [26], Qin et al. (2020) [27], Ren et al. (2022) [28], Salvi et al. (2022) [29], Salvi et al. (2024) [30], Sharafudeen et al. (2023) [31], Jütte et al. (2024) [32], Veeramani and Jayaraman (2025) [33], Yi et al. (2019) [8], Zou et al. (2024) [34].
Ethical or regulatory studies	3	Albisua (2025) [35], Bleher and Braun (2023) [36], Rezk et al. (2022) [37]
Academic papers (FDP/MT)	2	Higuera González (2025) [38], Redondo Hernández (2020) [39]
Others (comments, proposals, hybrids not focused on GANs)	4	Puri et al. (2022) [40], Karras et al. (2018) [41], Mirza and Osindero (2014) [42], Paladugu et al. (2023) [43].

**Table 2 bioengineering-12-01113-t002:** Methodological quality assessment of studies focused on the use of GANs in dermatology.

Article	Type of Study	Objective	Method	Level of Evidence	Strengths	Limitations
Alshardan et al. (2025) [16]	Methodological study	Improve analysis of unbalanced dermoscopic images	DCGAN to generate synthetic images; evaluation with ResNet50	III	Good performance with minority classes	Use of only one dataset, limited external validation
Behara et al. (2023) [17]	Methodological study	Synthesis and classification of skin lesions using an improved DCGAN	DCGAN with modified architecture + CNN classifier	III	Good classification performance after augmentation, testing with multiple lesion types	Limited dataset, no external validation
Carrasco Limeros et al. (2022) [18]	Comparative study	Evaluate different GAN models in dermatology	Comparison of StyleGAN and CycleGAN architectures	III	Extensive comparison, use of quantitative and qualitative metrics	Preprint study, lacking peer review
Ding et al. (2021) [19]	Methodological study	High-resolution dermoscopic image synthesis	cGAN with improved architecture for dermoscopic imaging	III	High image quality generated, robust metrics	Evaluation limited to synthetic quality, not clinical diagnosis
Ghorbani et al. (2020) [22]	Methodological study	Synthetic generation of clinical skin images with pathology	DermGAN (customized GAN with regularization to maintain pathologic features)	III	Preservation of relevant clinical features, morphological diversity, educational potential	Not clinically validated, dataset limited to Stanford images
Goceri (2024) [23]	Methodological study	Augmentation of hybrid lossy images	GAN with combined loss function (pixel + perception)	III	Improved class balance, direct applicability to classifier models	Limited to a single GAN type and database
Innani et al. (2023) [24]	Methodological study	Segmentation of lesions with GAN	GAN personalized with encoder–decoder architecture	III	Better delineation accuracy than U-Net	No comparison with other models
Heenaye-Mamode Khan et al. (2022) [25]	Methodological study	Multiclass classification of skin diseases	DGAN (modified GAN) combined with CNN	III	Good multiclass performance, complete metrics	External validation not performed
Karras et al. (2018) [41]	Foundational study	Improve quality and stability of GAN	Progressive GAN proposal with progressive training	IV	High synthetic quality, significant methodological impact	Not focused on dermatology
Mirza and Osindero (2014) [42]	Foundational study	Introduce the concept of conditional GAN	Conditional GAN with class vector	IV	Cornerstone for cGANs	No direct application in health
Paladugu et al. (2023) [43]	Methodological review	Review of the use of cGANs in medicine	Narrative review with classification of applications	V	Broad topic coverage, useful for developers	Does not evaluate empirical evidence
Perez and Ventura (2023) [26]	Methodological study	Data augmentation for skin cancer diagnosis	Progressive HLG for dermoscopic image generation	III	Improved diagnostic accuracy using synthetic images	Simulation-based assessment alone
Qin et al. (2020) [27]	Methodological study	Synthetic imaging classification of skin lesions	GAN for generation + CNN for classification	III	Improved sensitivity in classes with less data	Limited data and no clinical validation
Ren et al. (2022) [28]	Methodological study	Controlled generation of medical images	GAN with attribute control	III	Explicit control of variables in synthetic imaging	Not exclusively focused on dermatology
Salvi et al. (2022) [29]	Methodological study	Color standardization in dermatological images	DermoCC-GAN for visual standardization	III	Improved comparability of images between devices	Lack of direct clinical impact assessed
Salvi et al. (2024) [30]	Methodological study	Color standardization in pathology and dermatology	GANs with style transfer networks	III	Cross-cutting application in digital medicine	Non-dermatology specific
Sharafudeen et al. (2023) [31]	Methodological study	Detection of false synthetic images (deepfakes)	Vision Transformer on images generated with CGAN	III	High precision, novel combination of technologies	Low sample size
Veeramani and Jayaraman (2025) [33]	Methodological study	High-resolution melanoma image reconstruction	MELIIGAN (enhanced GAN model with residual blocks)	III	Intricate details preserved, high objective metrics	Not tested in clinical setting
Yi et al. (2019) [8]	Narrative review	Review of the use of GANs in medical imaging	Narrative review with task analysis and architecture	V	Detailed overview, basis for future research	Does not evaluate clinical efficacy directly
Zou et al. (2024) [34]	Methodological study	Segmentation of skin lesions using GANs with semantic awareness	GANs with global and local care modules	III	Significant improvement in segmentation metrics, innovative architecture	Validation on only two databases, no comparison to non-GAN methods

**Table 3 bioengineering-12-01113-t003:** Examples of commercial dermatology applications using GAN-based technology (date of the commercial tool search: April 2025).

Clinical Task	How a GAN Helps	Clinical Benefit	Commercial Apps^®^ Examples
Data augmentation for AI	Generates realistic synthetic images of lesions	Improves performance of AI models even with little data	SkinVision, DermaSensor
Progression simulation	Visually predicts lesion progression	Assists in treatment planning and patient education	FotoFinder, 3Derm
Image reconstruction	Restores blurred or poorly illuminated images	Improves diagnosis at a distance or with suboptimal images	ModMed EMA, Skinive
Image standardization	Corrects illumination and contrast differences contrast	Facilitates tracking and comparison between consultations	DermoCC-GAN, DermEngine
Lesion segmentation	Distinguishes lesion from background	Allows for objective measurement of the affected area and clinical follow- up	SkinIO, SLSNet
AI-assisted classification	Improves datasets to train classifiers	Automated diagnostic support or in triage systems	SkinVision, DermaSensor
Medical education	Creates didactic images of rare cases	Enhances training of students and residents	Crisalix, TouchMD
Skin tone inclusion	Generates images of diverse skin tones	Reduces bias in AI, improves diagnostic fairness diagnosis	Fitzpatrick 17k, SkinIO
Patient tools	Simulates evolution/improvement with treatment	Increases therapeutic adherence and patient understanding	Crisalix, ModMed Patient Portal
Esthetic/surgical simulation	Visualize effects of surgeries or treatments	Improves communication and informed consent	Crisalix, MirrorMe3D
Clinical Apps	Basis for monitoring or self-monitoring	Chronic patient management and remote monitoring	Skinive, DermaSensor

**Table 4 bioengineering-12-01113-t004:** Metrics to evaluate GANs in medical imaging.

Metric	What It Measures	Advantages	Limitations	Application in Dermatology
FID	Distance between feature distributions (Inception)	Sensitive to quality and diversity, correlates with human perception	Depends on extraction model (Inception not trained on medical data)	Widely used in studies of GANs in skin, although not specific
SSIM	Perceptual structural similarity	Captures differences in luminance, contrast, and structure	Does not consider semantic content or general realism	Useful in direct comparison (e.g., reconstruction or super-resolution)
IS	Diversity and realism using classification	Easy to implement	Does not compare to real data, biased by base network	Less used in medicine, more in natural imaging
PSNR	Pixel-to-pixel error (based on noise)	Simple and known	Non-perceptual, very sensitive to small differences	Used in reconstruction or resolution enhancement tasks
MSE	Mean squared difference between images	Intuitive and mathematically simple	Very limited for perceptual images	Similar to PSNR, useful for tasks where ground truth is known
LPIPS	Distance between deep activations (perceptual)	High correlation with human perception, robust to small deformations	More computationally expensive	Well suited for comparing realism in synthetic lesions
PPL	Latent space smoothness	Evaluates interpolations in latent space (important in GANs)	More useful for model evaluation than for direct image comparison	Indirect, useful in GAN development (e.g., StyleGAN)
Precision/Recall (distributional)	Fidelity vs. diversity of sample generated	Allows evaluation of collapse and overfitting mode	More complex to calculate	Useful when looking for variety of injury types
Mode Score	Realism and class coverage	Evaluates diversity when there are clear labels	Requires class annotations	Applicable in annotated datasets (e.g., lesion types)
Dice/Jaccard	Spatial overlap between masks or structures	Widely used in medical imaging with segmentation	Only applicable if segmentations are present	Ideal if images with defined anatomical regions

**Table 5 bioengineering-12-01113-t005:** Regulatory landscape for GAN-based applications in dermatology: European Union vs. United States.

Domain	European Union (EU)	United States (U.S.)
Primary Regulation	AI Act (2024, pending full implementation); Medical Device Regulation (MDR, 2017/745)	FDA regulations under the Federal Food, Drug, and Cosmetic Act and Software as a Medical Device (SaMD) Framework
Regulatory Authority	European Commission; National Competent Authorities; European Medicines Agency (EMA)	Food and Drug Administration (FDA); Center for Devices and Radiological Health (CDRH)
AI-specific Framework	AI Act classifies AI systems by risk (unacceptable, high, limited, minimal); GANs used in diagnosis are high-risk	FDA does not yet have a specific AI law but follows guidance on Good Machine Learning Practice (GMLP) and SaMD categories
Classification Approach	Risk-based: GANs for clinical decision-making would be “high-risk” under AI Act and Class IIa or higher under MDR	Function-based: based on intended use, risk level, and degree of autonomy; GANs used in diagnosis likely fall under Class II or III
Validation and Safety	Strict requirements under MDR for clinical evaluation, safety, and post-market surveillance	Requires premarket clearance or approval (510 (k), De Novo, PMA), and validation through clinical evidence
Ethical Guidelines	European Ethics Guidelines for Trustworthy AI (2019); human oversight and transparency mandatory	AI-based tools must demonstrate safety, effectiveness, and transparency; bias mitigation is a growing focus (FDA AI/ML Action Plan)
Data Privacy and Governance	GDPR compliance required, including data minimization and patient consent	HIPAA compliance governs medical data use and patient privacy protection
Ongoing Initiatives	EU AI Liability Directive; European Health Data Space; EUDAMED platform	Digital Health Center of Excellence; AI/ML SaMD Action Plan; real-world performance monitoring frameworks

AI: Artificial Intelligence; GAN: Generative Adversarial Network; MDR: Medical Device Regulation; SaMD: Software as a Medical Device; FDA: Food and Drug Administration; CDRH: Center for Devices and Radiological Health; GMLP: Good Machine Learning Practice; GDPR: General Data Protection Regulation; HIPAA: Health Insurance Portability and Accountability Act; PMA: Premarket Approval; EUDAMED: European Database on Medical Devices.

## Data Availability

Not applicable.

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
