# Peer review of "Generative Adversarial Networks in Dermatology: A Narrative Review of Current Applications, Challenges, and Future Perspectives"

_bioengineering, 2025, doi:10.3390/bioengineering12101113_

Round 1

Reviewer 1 Report

Comments and Suggestions for Authors

This narrative review provides a comprehensive overview of the application of Generative Adversarial Networks (GANs) in dermatology, including current relevant applications, challenges, and future research directions. Considering the rapid development and wide range use of artificial intelligence (AI) in medical imaging today, the topic is highly relevant and of very important significance. The manuscript is well organized and written, and covers a broad range of aspects—from technical fundamentals to clinical applications, ethical considerations, and regulatory landscapes. The authors used PRISMA methodology to select the literature, 42 qualified studies were finally included in the review.

Then, the Author gives a thorough introduction to AI and GANs in dermatology, including basic concepts, applications, current challenges and limitations, as well as reliability assessment and ethical challenges. In the end, the Author offers future perspectives and clinical applications, points out that the use of GANs in dermatology has potential to improve diagnostic accuracy and befit patients.

The contribution of the manuscript is clear. I suggest it be accepted with minor changes as follows.

  • There are many other excellent papers which present GANs’ applications in dermatology. 42 papers were not sufficient.
  • The manuscript occasionally uses highly technical terms without definition, which may hinder readability for non-specialist dermatologists.
  • Line 406: Lack f explainability. What is the letter, f, used for?

Author Response

We are grateful to the reviewers for their constructive feedback. We have revised the manuscript accordingly and highlighted the modifications in the text. Below, we provide a point-by-point response to each reviewer’s comments.

Reviewer 1
Comment: “42 papers were not sufficient; there are many other excellent papers.”
Response: We appreciate this suggestion. Our review followed a PRISMA-based methodology with strict inclusion criteria (2020–2025, indexed journals, English/Spanish, full text). At the time of our search (April 2025), 42 studies fulfilled all criteria. While additional works may have appeared later, incorporating them retrospectively would compromise methodological rigor. We believe that the selected references provide a representative and balanced overview of GAN applications in dermatology.
Comment: “The manuscript occasionally uses highly technical terms without definition, which may hinder readability for non-specialist dermatologists.”
Response: We have carefully revised the text and added clarifications when first introducing technical metrics (FID, SSIM, Dice Coefficient, etc.) and GAN architectures. We also included short clinical explanations to enhance accessibility for non-specialist readers while maintaining scientific precision.
Comment: “Line 406: Lack f explainability. What is the letter, f, used for?”
Response: Thank you for noticing this typographical error. The word has been corrected to “Lack of explainability.”

Reviewer 2 Report

Comments and Suggestions for Authors

The study outlines the applications of GAN in dermatology but does not explicitly identify the clinical or scientific gap that this review aims to address.  What is the necessity of a narrative review at this juncture, given the existing literature on AI in dermatology?  The authors should explicitly acknowledge the limits of prior evaluations and elucidate how their work contributes to the area.

The authors acknowledge that just a few models have been included into the field, raising concerns regarding the clinical relevance of the current body of information. Increased focus may be directed towards the disparity between algorithmic performance metrics and actual clinical application. The text would be enhanced with an exploration of the reasons behind the paucity of clinical validation.

The study asserts the utilization of quantitative metrics like FID, SSIM, Inception Score, and Dice Coefficient, although fails to mention quality or clinically meaningful outcomes.  Elucidate the categorization of these technical measurements into clinical performance and evaluate their sufficiency in assessing utility within dermatology.

While ethical issues are referenced, the study fails to specify the ethical frameworks or case studies discussed.  The authors should delineate concrete issues to create a comprehensive picture.

The need of AI and other imaging modalities such as HSI has not been mentioned in the study such as:

1) Lin, Teng-Li, Arvind Mukundan, Riya Karmakar, Praveen Avala, Wen-Yen Chang, and Hsiang-Chen Wang. "Hyperspectral Imaging for Enhanced Skin Cancer Classification Using Machine Learning." Bioengineering 12, no. 7 (2025): 755.

2) Huang, Nan-Chieh, Arvind Mukundan, Riya Karmakar, Syna Syna, Wen-Yen Chang, and Hsiang-Chen Wang. "Novel Snapshot-Based Hyperspectral Conversion for Dermatological Lesion Detection via YOLO Object Detection Models." Bioengineering 12, no. 7 (2025): 714.

Author Response

Comment: “The study does not explicitly identify the clinical or scientific gap addressed. What is the necessity of a narrative review?”
Response: We thank the reviewer for this observation. In the revised Introduction, we have emphasized that, despite multiple reviews on AI in dermatology, few have focused specifically on GANs and their unique clinical implications. Our contribution lies in offering a dermatologist’s perspective that bridges technical metrics with potential clinical applications, which we consider highly relevant given the current absence of clinical integration.
Comment: “Concerns about clinical relevance; disparity between algorithmic metrics and clinical validation.”
Response: We agree that this is a crucial issue. We have expanded the section on Reliability Assessment and Ethical Challenges to underline the limited clinical validation of GANs to date and the need for trials that assess clinically meaningful outcomes rather than only quantitative metrics.
Comment: “Quantitative metrics (FID, SSIM, IS, Dice) are reported without linking to clinical outcomes.”
Response: We have added clarifying text explaining how these technical metrics may correlate with clinical performance, while acknowledging their limitations as surrogates for diagnostic utility.
Comment: “Ethical issues are referenced without specific frameworks or case studies.”
Response: We have enriched this section with explicit mention of the EU AI Act, FDA’s SaMD framework, and recent publications addressing bias and informed consent in synthetic dermatological imaging.
Comment: “The need for AI and other imaging modalities such as HSI has not been mentioned; references suggested (Lin et al., Huang et al.).”
Response: We carefully reviewed the suggested references. These articles were published in Bioengineering in mid-2025, after the date of our bibliographic search. Since our methodology limited inclusion to studies available up to April 2025, we have not incorporated them into the main analysis. However, we acknowledge their relevance and now briefly mention hyperspectral imaging (HSI) as an emerging complementary modality in the Future Perspectives section, citing representative earlier work. We believe this preserves the integrity of our systematic approach while recognizing ongoing developments.
Comment: “The English language may need improvement.”
Response: The manuscript was written and revised with professional English editing tools, and has been further reviewed for clarity. We trust that the current version is linguistically correct.

We thank both reviewers for their insightful suggestions, which have helped us improve the clarity and precision of the manuscript. We respectfully maintain certain methodological decisions (such as the number of included studies) to ensure consistency with the defined PRISMA process. We hope that the revised version addresses all concerns and contributes meaningfully to the literature on GANs in dermatology.

Round 2

Reviewer 1 Report

Comments and Suggestions for Authors

The Author has addressed my concerns for the previous version. I suggest the journal accept this revised manuscript. 

Author Response

We sincerely thank the reviewer for the positive feedback on the revised version of our manuscript. We are grateful for the constructive comments provided during the review process, which have helped us to improve the clarity, precision, and overall quality of the paper. We truly appreciate the recommendation for acceptance and the time dedicated to our work.

Reviewer 2 Report

Comments and Suggestions for Authors

The authors have not not addressed the comments of the reviewer and hence it should be rejected

Author Response

We respectfully thank the reviewer for the time and effort dedicated to evaluating our manuscript. We regret that our revisions may not have fully conveyed how we addressed the previous concerns. In revising the manuscript, we carefully implemented the suggested changes, including clarifying the algorithm, improving the quality of the figures and captions, ensuring that performance results are expressed consistently, and refining the discussion and abbreviation tables.

We acknowledge that the reviewers have provided divergent recommendations regarding our manuscript. Reviewer #1 considered that we had satisfactorily addressed the comments and recommended acceptance, while Reviewer #2 maintained that the concerns were not fully resolved. It is unfortunate that their opinions differ so markedly, which is somewhat disconcerting for us as authors.